# Porous supraparticle assembly through self-lubricating evaporating colloidal ouzo drops

Huanshu Tan [1], Sanghyuk Wooh [2], Hans-Jürgen Butt [3], Xuehua Zhang [4] & Detlef Lohse [1,5]

The assembly of colloidal particles from evaporating suspension drops is seen as a versatile route for the fabrication of supraparticles for various applications. However, drop contact line pining leads to uncontrolled shapes of the emerging supraparticles, hindering this technique. Here we report how the pinning problem can be overcome by self-lubrication. The colloidal particles are dispersed in ternary drops (water, ethanol, and anise-oil). As the ethanol evaporates, oil microdroplets form ('ouzo effect'). The oil microdroplets coalesce and form an oil ring at the contact line, levitating the evaporating colloidal drop ('self-lubrication'). Then the water evaporates, leaving behind a porous supraparticle, which easily detaches from the surface. The dispersed oil microdroplets act as templates, leading to multi-scale, fractal-like structures inside the supraparticle. Employing this method, we could produce a large number of supraparticles with tunable shapes and high porosity on hydrophobic surfaces.

[1] Physics of Fluids Group, Max-Planck-Center Twente for Complex Fluid Dynamics, Mesa+ Institute, and J. M. Burgers Centre for Fluid Dynamics, Department of Science and Technology, University of Twente, P.O. Box 2177500 AE Enschede, The Netherlands. [2] School of Chemical Engineering & Materials Science, Chung-Ang University, 84 Heukseok-ro, Dongjak-gu, Seoul 06974, Korea. [3] Max Planck Institute for Polymer Research, 55128 Mainz, Germany. [4] Department of Chemical and Materials Engineering, University of Alberta, Edmonton, Alberta T6G1H9, Canada. [5] Max Planck Institute for Dynamics and Self-Organization, Am Faßberg 17, 37077 Göttingen, Germany. Correspondence and requests for materials should be addressed to S.W. (email: woohsh@cau.ac.kr) or to X.Z. (email: xuehua.zhang@ualberta.ca) or to D.L. (email: d.lohse@utwente.nl)

Supraparticles refer to three-dimensional macroscopic structures by self-assembly of colloidal (micro-) nanoparticles[1–4]. Such particles have been identified as prominent candidates for a wide variety of modern applications like catalysis[5,6], catalytically active particles[7], adsorbents in environmental pollution management[8,9], diagnostics[10], chromatography[11], photonics[1,12], barcodes[13], biomedical delivery[10], and sensing[14,15]. Supraparticle fabrication by drying microlitre colloidal dispersion drops on surfaces has been extensively investigated in last few decades, because of its versatility, operability, energy-efficiency, and potential scalability[1,3,16]. Through controlling evaporation rates[4], adding electric or magnetic fields[3,17,18], adjusting pH or ionic strength of dispersion[19,20], or regulating the surface properties of (multi-) particles and substrate[6,21–24], the generated supraparticles can acquire many impressive features, including high surface-to-volume ratio, long-range order, and periodicity at mesoscale[1,5,12,25–27].

However, the strong adhesion between colloidal dispersion drops and surfaces hinders this promising technique. Evaporating colloidal drops normally suffer from a pinned contact line. As a consequence, capillary flows inside the drop arise and carry the colloidal particles to the edge of the drop, leading to a ring deposition, i.e., the so-called coffee ring effect[28]. Up to now, a feasible way to minimize the impact of the coffee ring effect on supraparticle synthesis is employing super liquid-repellant surfaces, where the colloidal drops can have a reduced initial contact area[5,21]. These special surfaces, however, are high-cost and fragile. Besides, it is difficult to achieve a complete detachment of the synthesized supraparticles from the surfaces. Another potential method to overcome the pinning effect is by partially submerging the colloidal drops on a lubricated oil layer on the substrate or by using a lubricant-impregnate surface, but the mutual attractions of the floating drops due to the liquid surface deformation and the effect of gravity, i.e., the so-called cheerios effect, lower the controllability of this method[18,29,30].

Ouzo is the Greek anise-flavored aperitif, mainly consisting of water, ethanol, and anise-oil. As recently found[31], in the evaporating ouzo drops phase separation occurs due to the preferred evaporation of ethanol and the resulting lower oil solubility (ouzo effect), preferentially at the contact line, where oil droplets first form. Inspired by this phenomenon, we will employ the so-called ouzo effect to prevent the pinning of the evaporating colloidal drops.

In this paper, we propose a reliable, robust and cost-efficient method for supraparticle fabrication by employing a ternary liquid with appropriately chosen mutual solubilities (ouzo solution) in evaporation-driven nanoparticle assembly, which enables us to produce highly porous supraparticles with tunable shapes on commonly used hydrophobic surfaces.

## Results

### Evaporation-driven nanoparticle self-assembly experiments.
The method is achieved by using the ternary liquid, here consisting of milli-Q water (39.75 vol%), ethanol (59.00 vol%), and a slight amount of trans-anethole (1.20 vol%) (ouzo solution), as the suspension medium of $TiO_2$ nanoparticles (0.05 vol%). We deposited a drop of 0.5 µL ouzo suspension on a hydrophobic trimethoxy(octadecyl)silane (OTMS)-glass surface. A camera recorded the drop evaporation from the side (Fig. 1a). During the drying, an oil ring emerged under the colloidal drop[31]. After that, the drop shrank on the surface without a pinning contact line. After the evaporations of first the ethanol and then the water, a supraparticle emerged (Supplementary Movie 1).

We perform a control experiment (Fig. 1b) by evaporating a water-ethanol-nanoparticle drop (no oil contained, i.e., a binary liquid), with the same proportion of water, ethanol, and nanoparticle on the same substrate. In this case, no self-lubricating oil ring forms, and the nanoparticles deposites on the surface with various deposition patterns[32,33]. In a second control experiment, we evaporate an ouzo drop without dispersed nanoparticles (Fig. 1c). It has the same characteristics during evaporation as the case with all ingredients in Fig. 1a. The comparison of these three cases demonstrates that the self-formed oil ring plays a crucial role in the reduction of the contact diameter (illustration Fig. 1d), which leads to the formation of a supraparticle (Fig. 1e, f). The oil ring lubricates the evaporating colloidal drop during the self-assembly of nanoparticles. Therefore we call this process self-lubrication.

### Self-lubrication.
We further study the dynamics of the self-lubrication process and the nanoparticles self-assembly with a laser scanning confocal microscope (Supplementary Movies 2 and 3). The formation of the oil ring was followed by conducting a series of horizontal scans ≈10 µm above the substrate. Perylene (for oil) and rhodamine 6G (for aqueous) were added into the solution to distinguish different phases: Blue, yellow, black, and red in the confocal images of Fig. 2 represent the aqueous solution, the phase-separated oil, the nanoparticles (clusters), and the substrate, respectively. Initially, the colloidal ouzo drop was dark due to the dispersion of high concentration nanoparticles (Fig. 2a). The blue color of the solution became visible once nanoparticles started aggregating (inset Fig. 2b). The nucleated oil microdroplets attach to nanoparticles (clusters) due to the preference of heterogeneous nucleation on surfaces as compared to homogeneous nucleation in the bulk of liquids. Next, after the microdroplet nucleation, further nanoparticles will attach to the oil-aqueous interface[34]. Meanwhile, the nucleated oil microdroplets on the surface coalesced into an oil ring at the drop edge, which prevented the nanoparticles (clusters) from accumulating at the air-oil-substrate contact line (red-yellow boundary line in Fig. 2b). Driven by evaporation, the colloidal drop contracted radially, and the oil ring was forced to slide inwards (Fig. 2c). The drop contraction leads to nanoparticles assembly into a three-dimensional structure. Here, surface tension prevails over gravity, as the small drops have small Bond number $Bo = \rho g L^2 / \sigma \sim 10^{-1} \ll 1$, where $\rho$ is the density of the drop solution (~1000 kg m$^{-3}$), $g$ the gravitational acceleration, $L$ the characteristic size of the drop (~0.5 mm), and $\sigma$ the water/trans-anethole interfacial tension (~24.2 mN m$^{-1}$)[35].

The shrinkage of the oil ring causes the levitation of the colloidal drop, and the final geometry of the supraparticle is sculpted. The ridge of the oil ring rounds the edge of the colloidal drop (Fig. 2c). The inner ridge of the oil ring acts as the lower half of the dynamical mold for the self-assembly of nanoparticles, while the liquid-air interface makes the upper half. Consequently, the developing supraparticle is shaped by the oil wetting ridge. Therefore, by adjusting the oil concentration in the mixture, which results into different sizes of the oil wetting ridge, we are able to obtain different configurations of the mold and thus the different morphologies of the generated supraparticles (illustrated in Fig. 2d, e).

### Tunable shapes and high porosity of the supraparticles.
We control the shape of the generated supraparticles by varying the ratio $k$ of the oil volume fraction $\chi_{oil}$ to the nanoparticle volume fraction $\chi_{NP}$ in the initial colloidal solution. The full parameter space is shown in Fig. 3a, giving quantitative information on the final geometry (Fig. 3b) and the porosity (Fig. 3c) of the supraparticles. The ethanol-to-water volume ratio is 3:2 and the black dashed lines in the parameter space represent different

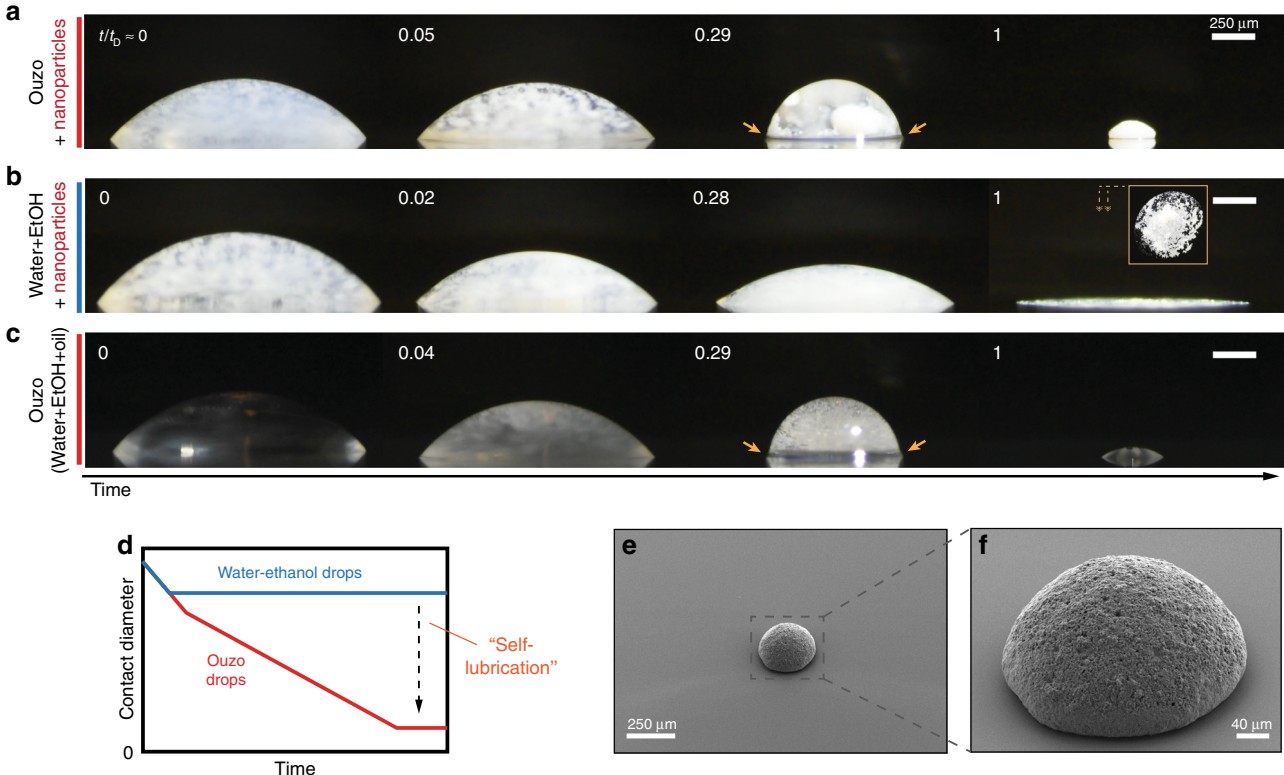

**Fig. 1** Supraparticles self-assembly through drying ouzo suspensions drops on hydrophobic surfaces. **a** Snapshots of the evaporation of a sessile drop of ouzo suspensions (water, ethanol, anethole oil, and nanoparticles). The contact diameter of the drop on surface smoothly decreased during the whole process because of the formation of an oil ring at the contact line (pointed at by arrows), and ultimately a supraparticle emerged (see below). The time $t$ is non-dimensionalized by the depletion time $t_D$. **b** The first control experiment by evaporating a sessile drop of water-ethanol suspensions with the same water-ethanol-nanoparticle proportion (without oil). The decrease of the contact diameter stopped soon and no supraparticle formed in the end. **c** The second control experiment by evaporating an ouzo drop with the same water-ethanol-anethole proportion (without nanoparticles), which displays the same dynamical evolution as in experiment **a**. The oil ring formed at the contact line of the drop is pointed at by an arrow. **d** A schematic illustration of the variation of contact diameter. In experiments **a** and **c** with the addition of the little anethole oil, the drops achieve a much smaller final contact diameter (red line) than in experiment **b** (blue line), which we call self-lubrication. **e** SEM photographs of the generated supraparticle from experiment **a**. **f** A close-up of the supraparticle. Scale bars in **a**-**c** are 250 μm

oil-to-nanoparticle ratios $\chi_{oil}/\chi_{NP}$. Each white square dot in Fig. 3a represents a solution composition used in experiments. The initial profile of the drop and the final profile of the supraparticle (after the depletion of oil) were captured by a gray-scale camera from the side, see Fig. 3d–g.

The experimental results reveal that the oil-to-nanoparticle ratio determines the supraparticle shapes. When the volume fraction of oil greatly exceeds the volume fraction of nanoparticles, a more spherical supraparticle forms (Fig. 3h). For less oil, the supraparticles assume more flat, oblate shapes (Fig. 3d–g). Although the oil wetting ridge and the configuration of the water-air-oil contact region determine the supraparticle shape, the nanoparticle aggregation and rearrangement during the supraparticle development have an effect on the final shape of the supraparticle as well. Data points a, b ($\chi_{oil} = 0$), and c ($\chi_{NP} = 0$) represent the oil and nanoparticle concentrations in the three cases displayed in Fig. 1a–c, respectively. If the amount of separated oil is not enough to form a complete oil ring, the repeatability of the supraparticle generation is not good (four data points in the grey region of Fig. 3a).

We define the geometric characteristics of the non-ball-like shape by the height and the depth of the dent of the oil ridge, i.e., $\delta h = H - h$ and $\delta l = l - L$ (annotations in Fig. 3e). We extracted this geometrical information through imaging analyses with a self-made MATLAB program, assuming axial-symmetry. The data in Fig. 3b show that both the dimensionless height $\delta h/h$ and

the dimensionless depth $\delta l/l$ increase monotonously with the increasing oil-to-nanoparticle ratio. The inset shows the dimensional data. The monotonic dependence reflects the fact that the oil wetting ridge sculpts the supraparticles. High oil ratios lead to a prominent oil wetting ridge, which causes a noticeable dent in the formed supraparticles.

Ball-like supraparticles are achievable when the oil-to-nanoparticle ratio is high enough to have the developing supraparticles submerged in the oil phase. A cohesive force of the interface layer between the surrounding oil and the colloidal drop pulled the developing supraparticle into spherical shapes. Thus ball-like supraparticles were generated, as displayed in the SEM image of Fig. 3h. The critical oil-to-nanoparticle ratio $k^*$ to have those ball-like supraparticles was estimated by a simple model. We assume a spherical-cap oil drop and a developing supraparticle submerged inside. Here, the developing supraparticle is in its upper limit size, which is equal to the height of the oil drop $H$, and the residual water fills the porous structure. With these assumptions, we have (see Methods section) $k^* = (3 \cot^2 \frac{\theta_{oil}}{2})/(1 - \phi)$, where $\phi$ is the porosity of the supraparticle, and $\theta_{oil}$ the contact angle of oil on the surface. Given 90% porosity and an advancing contact angle of 55° as obtained in our measurements, the calculated value is 110.7, corresponding to the red solid line in Fig. 3a, c. This line divides the parameter space into the white region of ball-like

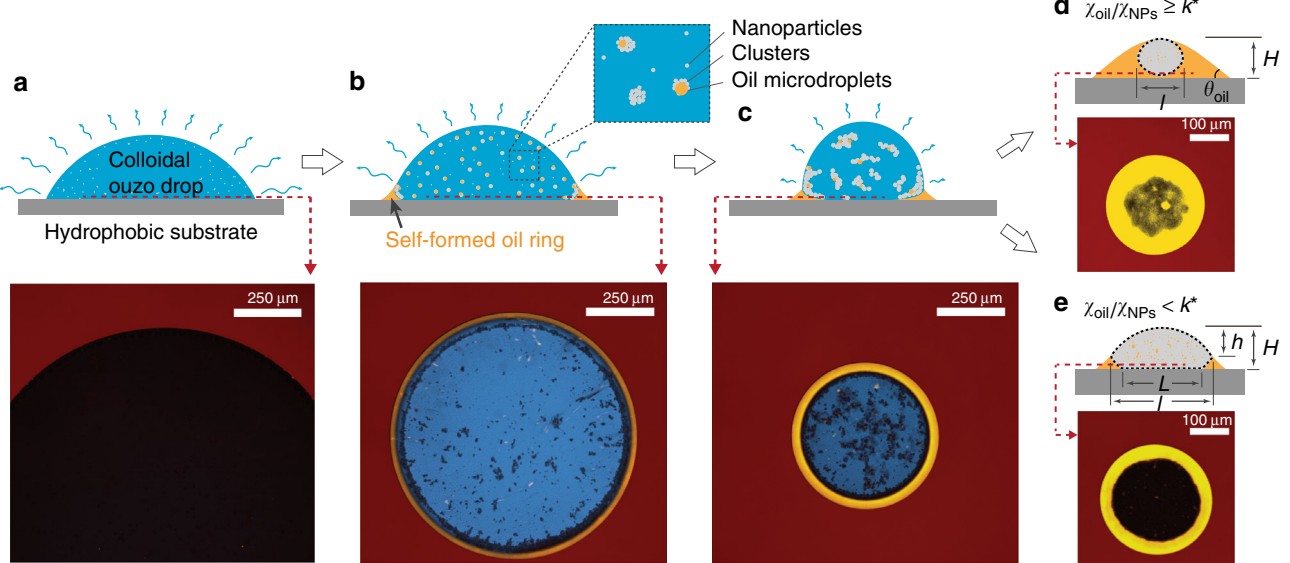

**Fig. 2** Illustrations of 'self-lubrication' and the corresponding confocal photographs. Color indications under a confocal microscope: yellow, oil; blue, water/ethanol; black, clusters of nanoparticles; red, substrate. **a** Initial state of evaporating drops of ouzo solution with well-dispersed nanoparticles. High nanoparticle concentration causes the black appearance of the drop under confocal. **b** Prevention of nanoparticle deposition at the contact line. Triggered by evaporation, the ouzo effect occurs, resulting in the formation of an oil ring (yellow), which prevents pinned contact lines and equips colloidal drops with high mobility and low hysteresis. Meanwhile, nanoparticles aggregate and oil microdroplets nucleate on them. **c** The shrinkage of the oil ring. The oil ring sweeps nanoparticles/clusters from the substrate. After the evaporation of ethanol and water, the generated supraparticles either float on the residual oil, as illustrated in **d**, or sit on the substrate, as illustrated in **e**, depending on the volume relationship between the supraparticle and the remaining oil. All the confocal photographs are from a horizontal scanning just above the substrate

supraparticles and the green region of supraparticles in diverse shapes, being consistent with our observations.

The obtained very high porosity of 90% and beyond is another prominent feature of the supraparticles. We calculated this porosity based on the initial volume of the colloidal drops, with the known nanoparticle concentrations and the final size of the supraparticles. The calculated porosity data shown in Fig. 3c range from 77 to 92% and monotonically increase with the oil-to-nanoparticle ratio. The nucleated oil microdroplets existing in the bulk of liquids provide a significant contribution to the porosity. Due to capillary forces, nanoparticle network forms among the nucleated oil microdroplets[34], which was also observed in our confocal image Fig. 2c, Supplementary Movies 2 and 3. After all the liquids (also oil) have diffused out, as a consequence, empty cells are left behind, dramatically increasing the porosity of the generated supraparticles. Increasing the oil-to-nanoparticle ratio increases the volume of those empty cells, so the porosity of the supraparticles rises (Fig. 3c). The limitation of the porosity (92%) is that, during the contraction of the developing supraparticle, the oil microdroplets merge up gradually and parts of them are absorbed into the oil ring[31].

The inner structure of the supraparticles verified the above explanation to the high porosity feature. To reveal this high porosity on all length scales in the interior of the supraparticle, we used the focused ion beam cutting technique (FIB) to investigate the supraparticle: Slide-by-slide cuts reveal the inner structure (Supplementary Movie 4). Figure 3i displays an exemplary cross-section of the supraparticle. It presents a multi-scale, fractal-like interior structure, and clearly shows that around half of the particle volume is made with micron-size holes (Fig. 3j). The rest contains many smaller holes of sub-micron size (Fig. 3k). Nanoparticles are joint together forming nanoparticle branches and mesopores (nanometer size) (Fig. 3l). These holes of (sub-)micron size originated from the nucleated oil microdroplets in the colloidal ouzo drop, as the nucleated oil microdroplets act as cells,

being devoid of (clusters of) nanoparticles during the supraparticle development (Supplementary Movie 5).

**Scalability of the supraparticle fabrication.** An engineering benefit of this method is the ease of scalability of the supraparticle fabrication. To give a demonstration to this advantage, we built a setup in our laboratory (Fig. 4a), which enables the automatic production of drops of similar size on trichloro(octadecyl)silane (OTS) or OTMS surfaces at rates of 20 drops per minute (Supplementary Movie 6). Few minutes after the drop being deposited, the supraparticles synthesis achieved. The supraparticle harvest was carried out by merely immersing the supraparticle-attached surface inside ethanol, and shaking them off with ease (Supplementary Movies 7 and 8). As a result, we had the supraparticles stored in liquid for future usage, and the surface was clean and ready for the next fabrication process. After several cycles, the supraparticle suspension was available. The self-lubricant layer and the complete detachment of the supraparticles increase the flexibility of supraparticle fabrication. Masses of supraparticles without controlled sizes could be fabricated through spraying the colloidal ouzo solution on the surface (Supplementary Movie 9).

By using different types of nanoparticles or multiple types of nanoparticles, we produced different kinds of metal oxide supraparticles for demonstration. Figure 4b–f is the SEM photographs of a large quantity of supraparticles generated by the self-assembly of $TiO_2$ nanoparticles (Fig. 4b), $TiO_2$ & $SiO_2$ nanoparticles (Fig. 4d), and $TiO_2$ & $SiO_2$ & $Fe_3O_4$ nanoparticles (Fig. 4f). Table 1 lists the composition of the ouzo solutions. Figure 4c displays the porous surface of the $TiO_2$ supraparticles. For the $TiO_2$ & $SiO_2$ supraparticles, the roughness difference is distinct at the top surface and the bottom surface (Fig. 4e). The calculated porosity is around 93%. Figure 4g, h is a sequence of zooms into the surface of the $TiO_2$ & $SiO_2$ & $Fe_3O_4$ supraparticle. The calculated porosity is around 91%. In Fig. 4h, different

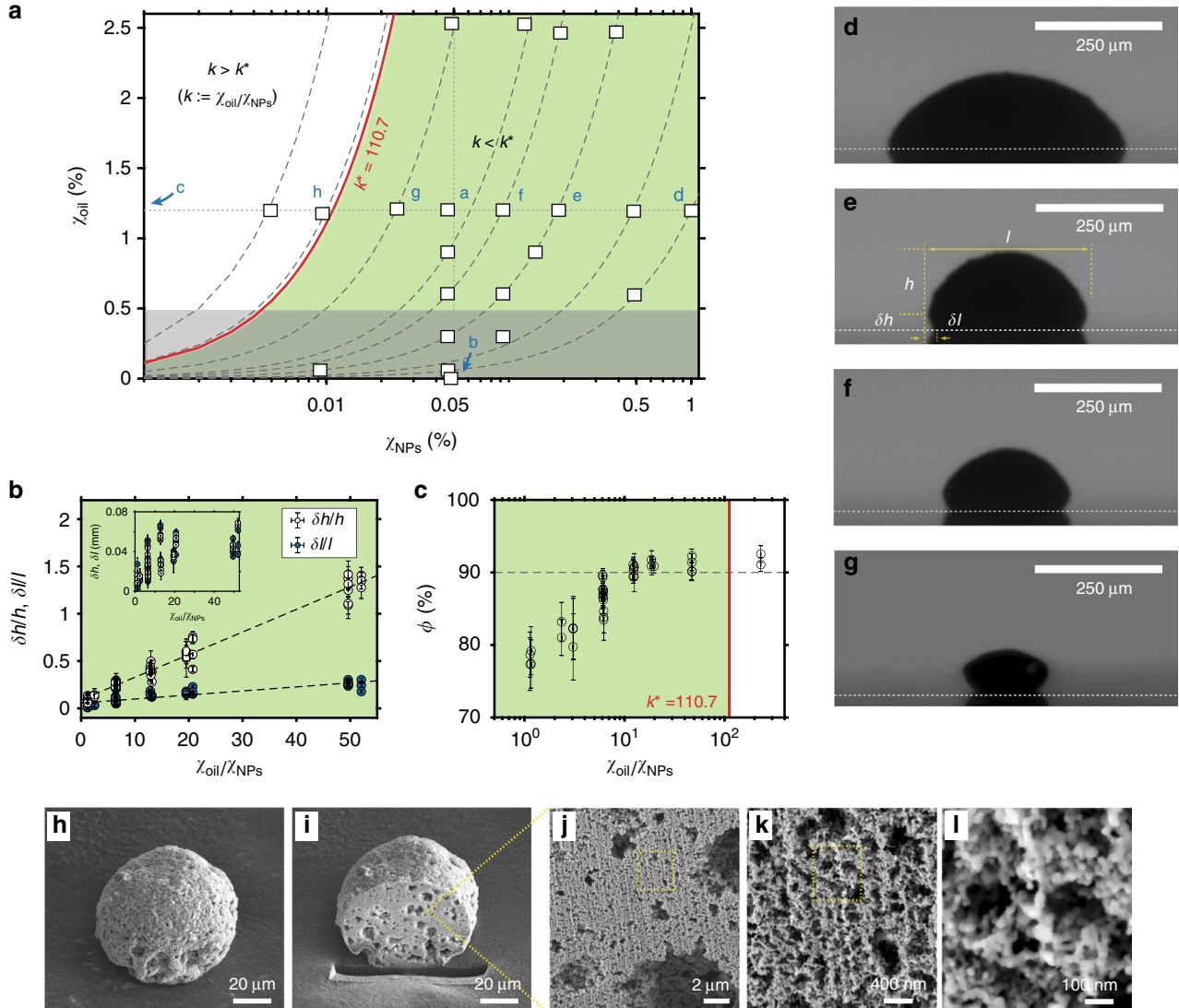

**Fig. 3** Supraparticles with tunable shapes and high porosity. **a** Parameter space showing the initial oil volume fraction $\chi_{oil}$ and nanoparticles volume fraction $\chi_{NP}$ of colloidal drops in different cases (white square dots) with a same ethanol-to-water ratio (3:2). The calculated critical oil-to-nanoparticle ratio, $k^* = 110.7$ (solid red line), divides the space into high ($k > k^*$) and low ($k < k^*$) oil-to-nanoparticle ratio regions. The generated supraparticles have a ball-like shape in the white region ($k > k^*$), while a more flat, oblate shape (see below) in the green region ($k < k^*$). **b** Both the dimensionless height $\delta h$ and depth $\delta l$ of the dented part of non-ball-like supraparticles are proportional to the oil-to-nanoparticle ratio in the green region. **c** The calculated porosity $\phi$ of supraparticles ranges from 78 to 92%. On increase of the oil-to-nanoparticle ratio, the diverse shapes are from a spherical-cap shape (profile photograph **d**), to a mushroom-like one **e**, **f**, and to a cupcake-like one **g**. Above the critical ratio $k^*$, a ball-like supraparticle is achievable (SEM image **h**). **i** Cross-section of the same supraparticle in **h** obtained by a FIB cutting illustrates the highly porous structure inside (Supplementary Movie 4). **j–l** A sequence of 3 zooms into the inner structure. The horizontal white dotted lines in **d–g** indicate the substrate position. The shadows below the lines are reflections. The image **e** shows the definitions of $\delta l$, $l$, $\delta h$, $h$. The error bars of the dimension and porosity of supraparticles represent the uncertainty in image processing. The error bars of the volume fraction of oil and nanoparticles represent the uncertainty from the preparation of the solution. The temperature and relative humidity during experiments were 20–23 °C and 35–50%, respectively

materials are distinguishable on the surface by virtue of an energy-selective backscatter detector (EsB): The bright spots pointed at by the yellow arrow are $Fe_3O_4$ nanoparticles; the light grey regions (the blue arrow) are $TiO_2$ nanoparticles; the dark grey regions (the red arrow) are $SiO_2$ nanoparticles. The darkness indicates holes on the surface.

## Discussion

In conclusion, our new method of mass production of self-lubricating, self-assembled supraparticles is a dramatic improvement over evaporation-driven supraparticle self-

assembly on super liquid-repellent surfaces and lubricant-impregnate surfaces. With our technique, commonly used planar hydrophobic surfaces are sufficient for the supraparticle fabrication, which improves flexibility, operability, and cost-efficiency of the fabrication. Besides that, the shapes of the generated supraparticle are tunable by varying the oil-to-nanoparticle ratio of colloidal solutions. Combing our method with piezoacoustic inkjet technology can dramatically scale-up fabrication, as there is no limitation on supraparticle collection. The generated highly porous supraparticles with multi-scale, fractal-like inner structures are suitable for many practical applications, such as catalysis, photonics,

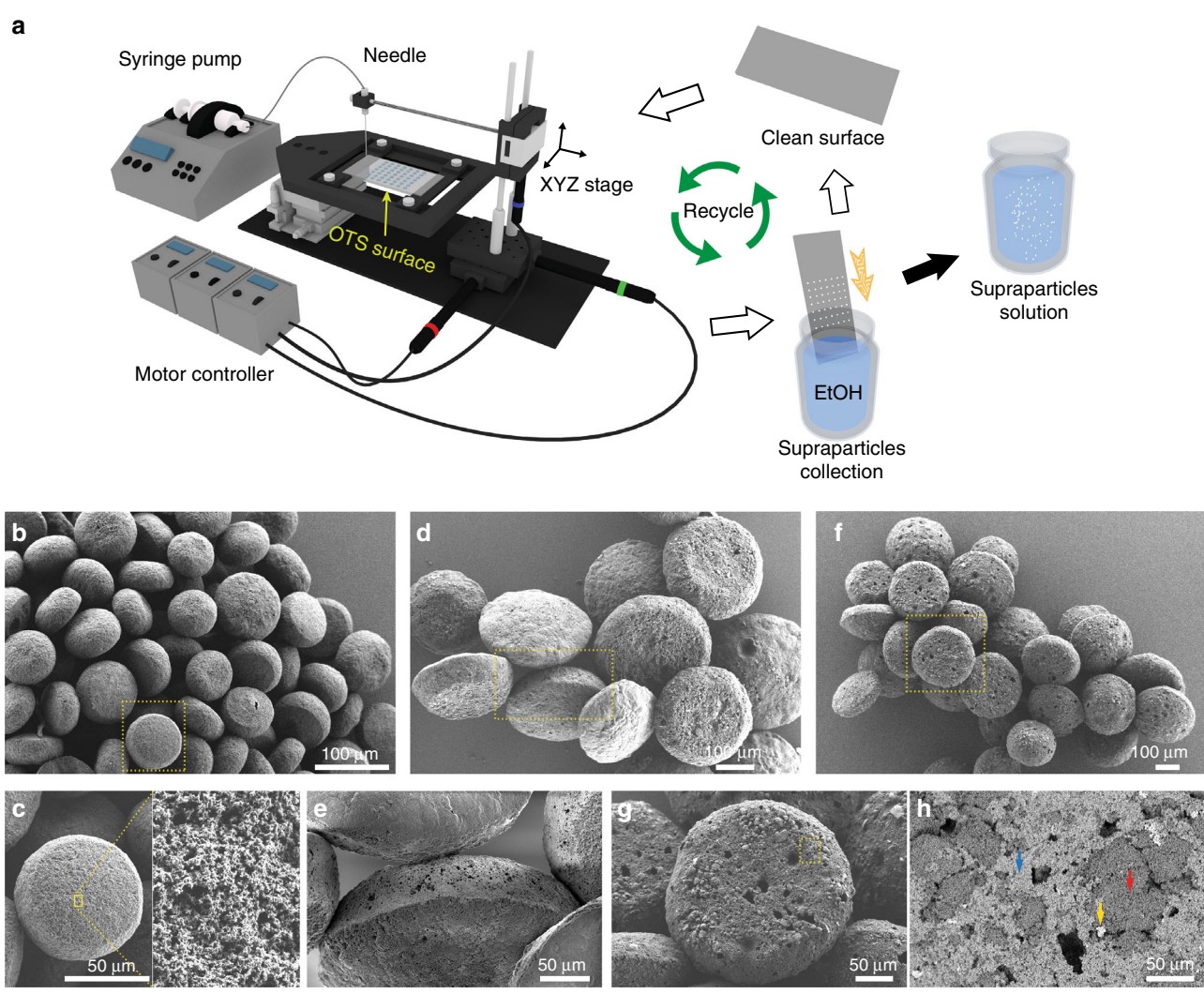

**Fig. 4** Scalability of the process with different and multiple types of nanoparticles. **a** Demonstration of the flexible and handy scalability of supraparticle fabrication on OTMS/OTS surface. As self-lubrication and the robust surfaces allow for a simple harvesting process and to recycle the surfaces. **b**–**h** SEM images of the generated supraparticles. **b** A large quantity of generated porous $TiO_2$ supraparticles. **c** A close-up view of the porous surface of the particle in **b**. **d** Bunches of porous supraparticles made by $TiO_2$ (0.05 vol%) and $SiO_2$ (0.05 vol%) nanoparticles. **e** A close-up of the side of the particle in **d**. **f** Bunches of porous supraparticles with three different nanoparticles, $TiO_2$ (0.06 vol%), $SiO_2$ (0.03 vol%), and $Fe_3O_4$ (0.01 vol%). **g**, **h** present a sequence of two zooms into the particle in **f**. In **h**, the supraparticle surface was imaged with energy-selective backscatter (EsB) detector to present different materials in different grey levels: $Fe_3O_4$ (bright spots pointed by the yellow arrow), $TiO_2$ (light grey regions by the blue arrow), $SiO_2$ (dark grey regions by the red arrow). Darkness indicates holes without nanoparticles

**Table 1 Composition list of the colloidal solutions for Fig. 4**

|  | Nanoparticles | | | Ouzo solution | | |
|---|---|---|---|---|---|---|
|  | TiO₂ | SiO₂ | Fe₃O₄ | Oil | Ethanol | Water |
| Fig. 4b | 0.005 vol% | — | — | 1.2 vol%[a] | 58.8 vol% | ~40 vol% |
| Fig. 4d | 0.05 vol% | 0.05 vol% | — | 1.8 vol%[b] | 58.2 vol% | ~40 vol% |
| Fig. 4f | 0.06 vol% | 0.03 vol% | 0.01 vol% | 1.7vol%[b] | 58.3 vol% | ~40 vol% |

[a]Anise oil (Sigma-Aldrich; Anise oil)
[b]Trans-anethole oil (Sigma-Aldrich; trans-anethole, ≥99.8%)

chromatography, environmental pollution management, and material science[1,5–9,11,12,14,15,36]. The self-lubrication effect, in conjunction with the easy detachment of the three-dimensional particle aggregates after the evaporation of particle-laden drops, implies a potential application in surface self-cleaning as well.

We also note that the nucleated oil microdroplets in the colloidal drop can act as a carrier phase for different purposes. Additionally, by controlling the composition and temperature of the ouzo solution, different morphological features of the nucleated oil microdroplets—size, number, distribution—are

tuneable as well[37]. Therefore, we expect more complex and exciting supraparticles created through this route.

## Methods

**Preparation of colloidal ouzo solution.** The purchased nanoparticles, titanium (IV) oxide (Aldrich, nanopowder, 21 nm, ≥99.5%), silicon dioxide (Aldrich, nanopowder, 10–20 nm, ≥99.5%), iron(II,III) oxide (Aldrich, nanopowder, 50–100 nm, 97%) were burned at 400 °C for 1 h to remove surfactants or contaminants attached on the particles before use. After that, the cleaned particles were added to specific amounts of Milli-Q water (produced by a Reference A+ system (Merck Millipore) at 18.2 MΩ cm at 25 °C) to make nanoparticle suspensions. Trans-anethole oil (Aldrich, 99%) and ethanol (Boom BV, 100% (v/v), technical grade) were used as received. Ethanol-oil (anethole) solutions were prepared separately beforehand and then mixed with the nanoparticle suspensions to make the final colloidal ouzo solutions with the required compositions for the experiments. We performed each mixing step in an ultrasonic bath for around 20 min.

**Preparation of the hydrophobic surfaces.** The chemicals used for the hydrophobic substrate preparation, trimethoxy(octadecyl)silane (Aldrich, 90%), toluene (Aldrich, 99.8%), tetrahydrofuran (Aldrich, ≥99.9%), and ethanol (Boom BV, 100% (v/v), technical grade) were used as received as well. In our experiments, the microscope glass slides (Thermo Scientific) were used as solid substrates for the octadecyltrimethoxysilane (OTMS) layer coating. We first carefully wiped the glass slides with ethanol wetted tissue for mechanically removing contaminants from the surfaces. Then the slides were successively sonicated in fresh acetone, ethanol, and Milli-Q water, each for 15 min, to remove organic contaminants from the surfaces. We repeated this step once and dried the slides by nitrogen flow. Then the slides were cleaned by plasma cleaner for 10 min. After that, the cleaned glass slides were immersed into the coating mixture of 1 vol% octadecyltrimethoxysilane and 99 vol % toluene for 3 h. After that, the coated slides were removed and then put into fresh toluene and tetrahydrofuran successively to dissolve the unlinked octadecyltrimethoxysilane above the surfaces. Finally, we dried the slides by nitrogen flow and put them in a clean Petri dish for temporary storage. The preparation of octadecylsilanes(OTS)-treated substrate follows the same process.

**The critical oil-to-nanoparticle ratio.** The initial oil volume fraction $\chi_{oil}$ and nanoparticle volume fraction $\chi_{NPs}$ are defined as $\chi_{oil} = V_{oil}/V_{all}$ and $\chi_{NPs} = V_{NPs}/V_{all}$, where $V_{oil}$, $V_{NPs}$, $V_{all}$ are the initial volumes of oil, nanoparticle, and solution, respectively. Thus, the initial oil-to-nanoparticles ratio $k$ is given by $k = \chi_{oil}/\chi_{NPs}$.

Here, we propose a simple model to estimate the critical oil-to-nanoparticle ratio $k^*$ of the colloidal suspension to acquire ball-like supraparticles. Illustration Fig. 2d shows a spherical supraparticle (SP) submerged in a spherical-cap oil drop. The maximum size of the supraparticle is the height of the oil drop $H$. From a simple geometrical consideration, we obtain the volume of the spherical supraparticle

$$V_{SP} = \frac{1}{6}\pi H^3,\tag{1}$$

and the volume of the oil drop (OD)

$$V_{OD} = \frac{1}{6}\pi H^3\left[1 + 3\cot^2\left(\frac{\theta_{oil}}{2}\right)\right],\tag{2}$$

with the oil contact angle $\theta_{oil}$. The volume of the oil is estimated as

$$V_{oil} = V_{OD} - V_{SP},\tag{3}$$

while the total volume of nanoparticles (NPs) is given by

$$V_{NPs} = V_{SP}(1 - \phi),\tag{4}$$

where $\phi$ is the porosity of the supraparticle. Thus, we obtain an estimation of the initial oil-to-nanoparticles ratio $k^*$ to have spherical supraparticles, namely

$$k^* = \frac{3}{1 - \phi}\cot^2\frac{\theta_{oil}}{2},\tag{5}$$

which only depends on the oil contact angle $\theta_{oil}$ and the supraparticle porosity $\phi$ and is independent of the drop size. In Eq. (3), we use $V_{SP}$ instead of $V_{NPs}$, because we assume that the residual water fills the porous structure.

## Data availability

The source data underlying Fig. 3a–c are provided as a Source Data file. The data that support the findings of this study are available from the authors upon reasonable request.

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

## Acknowledgements

We thank technicians Mark Smithers and Henk van Wolferen from NanoLab at University of Twente for the technological supports on scanning electro microscope. H.T. thanks for the financial support from the China Scholarship Council (CSC, file No. 201406890017). S.W. acknowledges his partner group in Max Planck Institute for Polymer Research. X.Z. acknowledges the support of the Natural Sciences and Engineering Research Council of Canada (NSERC) and Future Energy Systems (Canada First Research Excellence Fund). We also acknowledge the Netherlands Center for Multi-scale Catalytic Energy Conversion (MCEC) and the *Max Planck Center Twente for Complex Fluid Dynamics* for financial support.

## Author contributions

H.T. and S.W. designed research; H.T. performed research; H.T. analyzed data; H.T., S.W., H.B., X.Z. and D.L. discussed the results; H.T. and D.L. wrote the paper; H.T., S.W., H.B., X.Z. and D.L. revised the paper; Project supervision by X.Z. and D.L.

## Additional information

**Competing interests:** The authors declare no competing interests.

