## [Peer Review File · Nature Communications]

Reviewers' comments:

Reviewer #1 (Remarks to the Author):

Tan et al. report an interesting system, combining evaporation-driven formation of "supraparticles" with the "ouzo effect" leading to the formation of novel porous assemblies and their easy detachment due to the formation of an oil ring around the drying droplets. The report includes new combinations of effects, which are relevant to basic science and to materials fabrication and thus multidisciplinary. There are earlier studies on supraparticle assembly in the literature, but the present paper brings new elements that would be of broader interest and importance for publication in NC. The authors give generous credit to earlier work in the field, while introducing a new system that adds another phase, has new physical effects and leads to easier fabrication. Thus, the work is recommendable for publication. However, the authors should address various (relatively small) comments listed below:

My major concern for this work is that uses trans-anethole or anise oils as an essential ingredient. I am not sure of their cost and availability, but they could create serious problems, especially that the authors claim that the method is inexpensive. What other mixed liquid systems could be used for such processes? The authors should test and point out at least one system that has similar behavior based on more common chemical components.

P. 3: "the mutual attraction of the floating drops through the so-called 'Cheerios effect'". This effect is not necessarily known to the wider audience of the journal and could be confused with variations of coffee rings during drying. The authors should refer more physically to capillary forces between floating objects.

The authors refer to the self-formed oil ring illustrated in Fig. 1, but there is no visible ring in any of the experimental images there, which should be mentioned, or if there is a ring that can be noticed it should be pointed out.

Does the anise oil ring have a three-phase contact angle with the water/air surface (Fig. 2c), or form a continuous film on top of the water as hinted in Fig. 2d? These are two physically different cases and showing both of them make the figure and interpretation inconsistent. The authors explain their data based on relative volumes, but the O/W/A contact angle is likely constant and could be reported (or it should be interpreted if varies).

The data in Fig. 3 are collected at a variation of the relative humidity during experiments 35-50%. Such a variation is common in laboratory conditions but could lead to big variance in the evaporation rates too. Has this created larger data scattering and if yes/no, why?

The authors report the formation of various composite supraparticles containing TiO₂ & SiO₂ & Fe₃O₄ nanoparticles. These structures are not only interesting but could have various applications that could be listed. Do the supraparticles containing iron oxide display discernible magnetic properties?

Reviewer #2 (Remarks to the Author):

In this work the authors describe the formation of supraparticles obtained by evaporation from suspensions containing colloidal particles. This is in general an approach that has already been largely explored and has led to a large variety of interesting results. The really novel aspect of this work is the

usage of a pre-ouzo solvent for dispersing the particles. By evaporation of the most soluble component ethanol one moves into the ouzo regime, which means one forms small droplets of the contained oil anethole. The latter then forms a self-lubricating oil ring, which allows to control the shape of the finally formed supraparticles (after complete evaporation of all solvent), basically via the oil-to-nanoparticle ratio. The authors also nicely demonstrate that their approach is allowing to employ a number of different types of nanoparticles, as long as they are soluble in the aqueous phase. I consider this a very interesting and innovative approach of high degree of novelty in colloid science. In addition, it can be done on simply hydrophobic surfaces, which renders this process rather simple to do and is demonstrated to be nicely scalable by using the ink-jet technology.

There are only some minor points that I would recommend to be improved within a minor revision. These concern:

- at various places there are some inaccuracies with respect to the use of the English language. A careful reading with that respect should be done.

- on top of p. 6 it is stated "By changing the geometry of the oil wetting ridge, we could obtain different configuration of the mold". This is explained shortly after, but it might be good to make this connection of how exactly the oil wetting ridge is controlled somewhat clearer – especially for readers not so familiar with the different aspects of the system studied.

- in the legend of Fig. 3 it might be good to explain shortly but clearly the significance of the grey and green area (that comes later in detail, but it would facilitate fast reading if given here). Also in that legend shortly pointing out the porosity calculation would help in the same direction.

I enjoyed reading this manuscript very much and as already indicated would fully support publication in Nature Communications after the indicated minor revision.

Referee 1

Tan et al. report an interesting system, combining evaporation-driven formation of “supraparticles” with the “ouzo effect” leading to the formation of novel porous assemblies and their easy detachment due to the formation of an oil ring around the drying droplets. The report includes new combinations of effects, which are relevant to basic science and to materials fabrication and thus multidisciplinary. There are earlier studies on supraparticle assembly in the literature, but the present paper brings new elements that would be of broader interest and importance for publication in NC. The authors give generous credit to earlier work in the field, while introducing a new system that adds another phase, has new physical effects and leads to easier fabrication. Thus, the work is recommendable for publication. However, the authors should address various (relatively small) comments listed below:

Authors: We thank the referee for his/her reading of our manuscript, for pointing out the importance of our work, and for his/her recommendation for publication in Nature Communications. In the following we reply to the comments (reproduced in black) one by one (in blue) and explain how we revised the manuscript.

1. Referee: *My major concern for this work is that uses trans-anethole or anise oils as an essential ingredient. I am not sure of their cost and availability, but they could create serious problems, especially that the authors claim that the method is inexpensive. What other mixed liquid systems could be used for such processes? The authors should test and point out at least one system that has similar behavior based on more common chemical components.*

Authors: We thank referee 1 for pointing out this concern. Trans-anethole or anise oils are cheap and widely used as a flavoring substance in alcoholic drinks ouzo, rakı and Pernod (aperitifs that being widely consumed in Greece, Turkey, and France). Moreover, the oil ingredient used in this method is only a few percentages of total solution. Thus we indeed can claim that the method is

inexpensive. The essential part of this method is the “ouzo effect”, which lead to the nucleation of oil microdroplets and consequent formation of a self-lubricating oil ring. Thus this method is not restricted to anise oils. Many different other organic liquids can also be used in ouzo solution, as reported in previous works [1–4]. We now further stress the generality of the method in our revised manuscript.

2. Referee: *P. 3: “the mutual attraction of the floating drops through the so-called ‘Cheerios effect’”. This effect is not necessarily known to the wider audience of the journal and could be confused with variations of coffee rings during drying. The authors should refer more physically to capillary forces between floating objects.*

Authors: We agree with referee 1 and now added some physical explanation to the “Cheerios effect” in our revised manuscript.

Modification:

P. 3: ... but the mutual attraction of the floating drops due to the liquid surface deformation and the effect of gravity, i.e., the so-called “Cheerios effect”, lower the controllability of this method [5–7].

3. Referee: *The authors refer to the self-formed oil ring illustrated in Fig. 1, but there is no visible ring in any of the experimental images there, which should be mentioned, or if there is a ring that can be noticed it should be pointed out.*

Authors: Figure 1 in our revised manuscript is modified, and an arrow was added to point out the oil ring.

Modification:

The caption of Fig. 1: ... which we call “self-lubrication.” The oil ring formed at the contact line of the drop is pointed at by an arrow. The time ...

4. Referee: *Does the anise oil ring have a three-phase contact angle with the water/air surface (Fig. 2c), or form a continuous film on top of the water as hinted in Fig. 2d? These are two physically different cases and showing both of them make the figure and interpretation inconsistent. The authors explain their data based on relative volumes, but the O/W/A contact angle is likely constant and could be reported (or it should be interpreted if varies).*

Authors: There is a three-phase contact line among air, oil, and aqueous phases in our system as displayed in Figure 2c. Figure 2d and Figure 2e illustrate two different final stages after both ethanol and water have evaporated. The O/W/A contact angle is constant under ideal conditions without particles [8]. But during the nanoparticle aggregation or the formation of nanoparticle network in the later stage, the force equilibrium at the three phase contact line can be affected. Now we added this interpretation in our revised manuscript.

Modification:

Page 6: ... oblate shapes (Fig.3d-g). Although the oil wetting ridge and the configuration of

water-air-oil contact region determined the supraparticle shape, the nanoparticle aggregation and rearrangement during the supraparticle development have an effect on the final shape of the supraparticle as well. Data points a, b ...

5. Referee: *The data in Fig. 3 are collected at a variation of the relative humidity during experiments 35-50%. Such a variation is common in laboratory conditions but could lead to big variance in the evaporation rates too. Has this created larger data scattering and if yes/no, why?*

Authors: We agree with referee 1 that the relative humidity has an influence on the evaporation process. There are two distinct evaporation stages for evaporating ouzo drops [8–10]. In the first stage the evaporation is dominated by the ethanol evaporation, which indicates that the evaporation process is hardly affected by the relative humidity [8–10]. Only during the second stage, when only water evaporates, the relative humidity starts to play a role. However, how the evaporation rate affects the porosity of the final structure in our case is still not known, and a systematic investigation is required.

6. Referee: *The authors report the formation of various composite supraparticles containing TiO₂ & SiO₂ & Fe₃O₄ nanoparticles. These structures are not only interesting but could have various applications that could be listed. Do the supraparticles containing iron oxide display discernible magnetic properties?*

Authors: We fully agree with referee 1 that these structures could have various potential applications, including some listed in our manuscript. For demonstration purposes, we produced different kinds of supraparticles. We thank the referee for pointing out this interesting question. The properties of different generated supraparticles clearly deserve further investigations.

References

- [1] Steven A Vitale and Joseph L Katz. Liquid droplet dispersions formed by homogeneous liquid- liquid nucleation: “the ouzo effect”. *Langmuir*, 19(10):4105–4110, 2003.
- [2] Xuehua Zhang, Ziyang Lu, Huanshu Tan, Lei Bao, Yinghe He, Chao Sun, and Detlef Lohse. Formation of surface nanodroplets under controlled flow conditions. *Proceedings of the National Academy of Sciences*, 112(30):9253–9257, 2015.
- [3] Ziyang Lu, Amgad Rezk, Fernando Jativa, Leslie Yeo, and Xuehua Zhang. Dissolution dynamics of a suspension droplet in a binary solution for controlled nanoparticle assembly. *Nanoscale*, 9(36):13441–13448, 2017.
- [4] Ziyang Lu, Martin H Klein Schaarsberg, Xiaojue Zhu, Leslie Y Yeo, Detlef Lohse, and Xuehua Zhang. Universal nanodroplet branches from confining the ouzo effect. *Proceedings of the National Academy of Sciences*, 114(39):10332–10337, 2017.
- [5] Jeffrey R Millman, Ketan H Bhatt, Brian G Prevo, and Orlin D Velev. Anisotropic particle synthesis in dielectrophoretically controlled microdroplet reactors. *Nature Materials*, 4(1):98, 2005.
- [6] Stefan Karpitschka, Anupam Pandey, Luuk A Lubbers, Joost H Weijts, Lorenzo Botto, Siddhartha Das, Bruno Andreotti, and Jacco H Snoeijer. Liquid drops attract or repel by the inverted cheerios effect. *Proceedings of the National Academy of Sciences*, 113(27):7403–7407, 2016.
- [7] Dominic Vella and L Mahadevan. The “cheerios effect”. *American Journal of Physics*, 73(9):817–825, 2005.
- [8] Huanshu Tan, Christian Diddens, Pengyu Lv, J. G. M. Kuerten, Xuehua Zhang, and Detlef Lohse. Evaporation-triggered microdroplet nucleation and the four life phases of an evaporating ouzo drop. *Proceedings of the National Academy of Sciences*, 113(31):8642–8647, 2016.
- [9] Christian Diddens, Huanshu Tan, Pengyu Lv, Michel Versluis, JGM Kuerten, Xuehua Zhang, and Detlef Lohse. Evaporating pure, binary and ternary droplets: thermal effects and axial symmetry breaking. *Journal of Fluid Mechanics*, 823:470–497, 2017.
- [10] Huanshu Tan, Christian Diddens, Michel Versluis, Hans-Jurgen Butt, Detlef Lohse, and Xuehua Zhang. Self-wrapping of an ouzo drop induced by evaporation on a superamphiphobic surface. *Soft Matter*, 13:2749–2759, 2017.

Referee 2

In this work the authors describe the formation of supraparticles obtained by evaporation from suspensions containing colloidal particles. This is in general an approach that has already been largely explored and has led to a large variety of interesting results. The really novel aspect of this work is the usage of a pre-ouzo solvent for dispersing the particles. By evaporation of the most soluble component ethanol one moves into the ouzo regime, which means one forms small droplets of the contained oil anethole. The latter then forms a self-lubricating oil ring, which allows to control the shape of the finally formed supraparticles (after complete evaporation of all solvent), basically via the oil-to-nanoparticle ratio. The authors also nicely demonstrate that their approach is allowing to employ a number of different types of nanoparticles, as long as they are soluble in the aqueous phase. I consider this a very interesting and innovative approach of high degree of novelty in colloid science. In addition, it can be done on simply hydrophobic surfaces, which renders this process rather simple to do and is demonstrated to be nicely scalable by using the ink-jet technology. There are only some minor points that I would recommend to be improved within a minor revision.

.....

I enjoyed reading this manuscript very much and as already indicated would fully support publication in Nature Communications after the indicated minor revision.

Authors: We thank referee 2 for his/her positive assessments and for pointing out the high degree of novelty of our work. We appreciate that he/she “enjoyed reading this manuscript very much and as already indicated would fully support publication in Nature Communications after the indicated minor revision.”

1. Referee: at various places there are some inaccuracies with respect to the use of the English language. A careful reading with that respect should be done.

Authors: We have carefully read the manuscript once more and modified the inaccuracies.

2. Referee: *on top of p. 6 it is stated “By changing the geometry of the oil wetting ridge, we could obtain different configuration of the mold”. This is explained shortly after, but it might be good to make this connection of how exactly the oil wetting ridge is controlled somewhat clearer – especially for readers not so familiar with the different aspects of the system studied.*

Authors: We thank referee 2 for his/her careful reading. We agree with referee 2 and now added a connection in our revised manuscript.

Modification:

Page 6: ... Consequently, the developing supraparticle was shaped by the oil wetting ridge. Therefore, by varying the oil concentration in the mixture, which results into different sizes of the oil wetting ridge, we could obtain different configurations of the mold and thus the different morphologies of the generated supraparticles (Fig. 2de).

3. Referee: *in the legend of Fig. 3 it might be good to explain shortly but clearly the significance of the grey and green area (that comes later in detail, but it would facilitate fast reading if given here). Also in that legend shortly pointing out the porosity calculation would help in the same direction.*

Authors: We agree with referee 2. A new Figure 3 and the corresponding caption are in our revised manuscript.

Modification:

Caption of Fig.3: ... a same ethanol-to-water ratio (3:2). The calculated critical oil-to-nanoparticle ratio, $k^* = 110.7$ (red solid line), divides the space into high (grey) and low (green) oil-to-nanoparticle ratio regions. The generated supraparticles have a ball-like shape (SEM image **h**) in the grey region, while a more flat, oblate shape (profile photographs **d-g**) in the green region. **b**, Both the dimensionless height and depth of the dented part (defined in panel **e**) of non-ball-like supraparticles are proportional to the oil-to-nanoparticle ratio in the green region. On increase of the ratio, the diverse shapes are from a spherical-cap shape (**d**), to a mushroom-like one (**e,f**), and to a cupcake-like one (**g**). **c**, the calculated porosity

REVIEWERS' COMMENTS:

Reviewer #1 (Remarks to the Author):

I have examined the responses of the authors and the revised manuscript. I believe my comments have been taken into account or responded appropriately. The same applies to the comments of the other reviewer. Notably, both reviews are rather favorable. Thus, I recommend publication without further changes. Signed Orlin D Velez.

Reviewer #2 (Remarks to the Author):

In my opinion this is a very important contribution to interface and colloid science that with these modifications done should be published.

All points of criticism raised have been answered in a convincing manner.

Referee 1

I have examined the responses of the authors and the revised manuscript. I believe my comments have been taken into account or responded appropriately. The same applies to the comments of the other reviewer. Notably, both reviews are rather favorable. Thus, I recommend publication without further changes. Signed Orlin D Velev.

Authors: We thank Prof. Dr. Orlin D. Velev for examining our responses and revised manuscript. We appreciate that he recommends publication without further changes. We also thank him and the other reviewer for their comments to help us improve our manuscript.

Referee 2

In my opinion this is a very important contribution to interface and colloid science that with these modifications done should be published. All points of criticism raised have been answered in a convincing manner.

Authors: We thank referee 2 for his/her reading our revised manuscript and responses. We appreciate that he/she suggests the publication of our revised manuscript. We are grateful that he/she points out the very important contribution of our work to interface and colloid science. We also thank him/her for his/her comments to help us improve our manuscript.